# Effects of Ultra-Processed Diets on Adiposity, Gut Barrier Integrity, Inflammation, and Microbiota in Male and Female Mice

**DOI:** 10.3390/nu17193116

**Published:** 2025-09-30

**Authors:** Caroline de Menezes, Clara Machado Campolim, Angie Triana, Kênia Moreno de Oliveira, Leticia Gama S. Calixto, Fernanda Garofalo Xavier, Mario J. A. Saad, Everardo Magalhães Carneiro, Patricia O. Prada

**Affiliations:** 1School of Applied Sciences, State University of Campinas, Limeira 13484-350, SP, Brazil; carolinemenezes50@yahoo.com (C.d.M.); angieetriana@gmail.com (A.T.); 2Department of Internal Medicine, School of Medical Science, State University of Campinas, Campinas 13083-887, SP, Brazil; claracampolim@gmail.com (C.M.C.); leeticia.gama@gmail.com (L.G.S.C.); msaad@unicamp.br (M.J.A.S.); 3Obesity and Comorbidities Research Center (OCRC), Department of Structural and Functional Biology, Institute of Biology (IB), State University of Campinas, Campinas 13083-864, SP, Brazil; keniamorenoo@hotmail.com (K.M.d.O.); fernandax04@gmail.com (F.G.X.); emc@unicamp.br (E.M.C.)

**Keywords:** ultra-processed foods, intestine barrier, gut microbiota, inflammation, cytokines, obesity, feeding

## Abstract

Background/Objectives: The consumption of highly palatable ultra-processed foods (UPFs), enriched in sugar, saturated fat, and salt, increases the risk of morbidity and mortality by inducing obesity, type 2 diabetes (T2DM), cardiovascular disease, and cancer. The present study aimed to investigate the impact of a UPF-rich diet on adiposity, feeding behavior, glucose homeostasis, intestinal barrier markers, expression of inflammatory cytokines, and microbiota in male and female C57BL/6J mice. Methods: Animals received a chow diet or a UPF diet for 10 (UPF10) or 30 days (UPF30). UPF10 induced greater calorie intake as early as 10 days on a UPF diet. Fat accumulation occurs in both sexes, specifically after 30 days of exposure. Results: The duration of UPF exposure significantly influenced glucose metabolism and insulin sensitivity. A 10-day UPF diet was associated with lower fasting blood glucose levels, without higher insulin levels, in both sexes. Females showed early impairment in glucose tolerance. Male mice on UPF30 exhibited elevated systemic IL-6 levels, as well as reduced intestinal expression of *Occludin* and *E-cadherin* genes. In females, UPF30 increased *TNF-α* expression in the gut and increased microbial diversity. Both sexes displayed dysbiosis, with females showing pronounced changes in the proportion between predominant phyla, and males showing more specific changes in bacterial genera. Conclusions: A diet high in UPFs promoted metabolic, inflammatory, and gut microbiota alterations, with effects varying according to exposure duration and biological context, and becoming more pronounced after 30 days.

## 1. Introduction

The consumption of highly palatable ultra-processed foods (UPFs), enriched in sugar, saturated fat, and salt, increases the risk of morbidity and mortality by inducing obesity, type 2 diabetes (T2DM), cardiovascular disease, and cancer. Epidemiological data indicate that individuals with high UPF consumption have a 62% increased risk of all-cause mortality, and each additional serving of UPF raises this risk by 18% [1]. UPF intake is also associated with a higher risk of bowel cancer, irritable bowel syndrome, functional dyspepsia, and depression [2,3,4,5].

Ultra-processed foods (UPFs) are products manufactured by the food industry, typically containing high levels of sugar, salt, saturated fats, refined starches, syrups, and protein isolates. They are almost devoid of fiber, vitamins, and micronutrients [6]. Unlike natural foods, which do not usually contain significant amounts of both fat and sugar simultaneously, the food industry often combines these nutrients in UPFs [7]. The combination of fat and sugar can evoke effects similar to those of addictive drugs [8], promoting excessive consumption [7,9]. Hyperphagia triggered by excessive UPF intake may result from disruption of normal nutrient-sensing pathways in the gut and brain, potentially enhancing their reinforcing effects [10].

The gut–brain axis is a regulatory network that controls food intake and satiety, helping to maintain energy balance [11]. This axis involves bidirectional communication between the nervous system and the gastrointestinal tract, incorporating components such as the microbiome, the enteric nervous system, and gut hormones [12,13,14].

In addition to performing functions related to digestion and nutrient absorption, the intestine harbors the intestinal microbiota, which directly influences health by modulating the immune system and metabolism [15]. Therefore, it functions as a crucial interface for microorganisms, linking the internal and external environments [16].

Junctional complexes, known as adherens junctions (AJs) and tight junctions (TJs), connect and polarize epithelial cells, forming a physical barrier that segregates the microbiota and prevents pathogens from entry. The transmembrane protein E-cadherin is a key component of AJs, located at the border of the apical membrane of epithelial cells [17]. It promotes cell–cell adhesion and contributes to maintaining epithelial integrity. Besides delimiting the cellular space [16], AJs regulate leukocyte passage, facilitate water and electrolyte absorption, and render the space inaccessible to microorganisms [18]. AJ proteins likewise participate in the induction of TJ protein expression [19]. Occludin, a critical component of TJs, plays a vital role in regulating the intestinal barrier by controlling the movement of proteins, lipids, and ions through the paracellular pathway [20]. A reduction in occludin expression due to TJ disruption increases intestinal permeability, allowing harmful substances to enter the bloodstream and potentially triggering inflammation and disease [21].

The intestinal microbiota is essential for balanced immune responses [22] and for maintaining the integrity of the intestinal barrier by supporting TJ stability. Previous studies have shown that dysbiosis is often associated with increased permeability and inflammation of the digestive tract, as well as elevated bacterial translocation into the systemic circulation, which can lead to obesity and T2DM [16,22,23].

The UPF diet has been used in experimental animal models since the 1970s, first described by Sclafani and Springer to induce obesity, metabolic alterations, and changes in feeding behavior [24]. To provide precise compositions of proteins, minerals, and vitamins, the industry developed commercially available animal diets, such as the Western diet in pellet form (e.g., Research Diets D12079B or Envigo/Teklad: TD.88137) [14,15]. However, these pellets do not replicate natural feeding behaviors and can lead to decreased caloric intake in rodents. This reduction has been attributed to a lack of sensory variety in the pellets [25], suggesting that flavor, texture, and novelty are essential for stimulating food intake [26,27]. Over time, our group and others have developed palatable diets using industrialized foods, which are recognized as valuable tools for investigating insulin resistance, hyperphagia, obesity, low-grade inflammation, and accumulation of abdominal fat in rodents [28,29]. These diets typically include several items classified as UPFs according to the NOVA classification [30].

In the present study, we employed a similar UPF-enriched diet, previously used by our group [28]. Our results demonstrated that the UPF diet induced hyperphagia, increased adiposity, and triggered a pre-diabetic state in mice. In addition, we investigated the effects on the intestinal barrier and microbiota diversity, shedding light on the complex interactions between diet, energy metabolism, and sex differences.

## 2. Materials and Methods

### 2.1. Animals

The Care of Animals and Ethical Committee for Animal Research of the State University of Campinas approved all animal experiments and handling performed in the present study under the number CEUA Protocol 5853-1/2021. After the approval, the Central Animal Facility of UNICAMP CEMIB/UNICAMP provided eight-week-old male and female C57BL/6J mice. The animals were maintained at a temperature of 22 °C, with a fixed light-dark cycle (12/12 h), and received food and water ad libitum.

### 2.2. Diets

In the present study, we used a diet enriched with ultra-processed foods (UPF), based on a Western diet described before [28]. Items included in our UPF diet, such as milk chocolate, cornstarch cookies, industrialized pineapple-flavored cake, bacon-flavored snacks, and a soft drink, are all recognized as UPF according to the NOVA classification [30].

Briefly, the UPF diet consisted of standard Nuvilab CR1 chow (37.5%), peanuts (25%), milk chocolate (25%), and cornstarch cookies (12.5%). We crushed and minced the ingredients together and shaped the pellets [28]. In addition to the pellets, mice received industrialized pineapple-flavored cake made from refined wheat flour, industrialized bacon-flavored snacks, and a soft drink after gas removal (6 h in a magnetic stirrer at room temperature). We offered all food and soft drinks ad libitum. This diet provided 4.6 kcal per gram, comprising 50.3% carbohydrates, 14.6% protein, and 25% fat.

In contrast, the control group received a standard Nuvilab CR1 pellet, referred to as Chow Diet (CD), containing 53% carbohydrates, 22% proteins, and 4% lipids, with filtered water available ad libitum. The CD provided approximately 3.4 kcal per gram. The dietary composition is included (Appendix A), presenting the macro- and micronutrient content of each ultra-processed food (UPF) component used in the diet, based on data from the IBGE and TACO Brazilian Food Composition Tables.

We randomly divided the animals into four groups based on the diet and drink they received.

CD10: C57BL/6J mice receiving a chow diet (CD) and filtered water ad libitum for 10 days, independently of sex.

UPF10: C57BL/6J mice receiving a diet enriched with ultra-processed foods (UPF) and a soft drink without gas ad libitum for 10 days, independently of sex.

CD30: C57BL/6J mice receiving a chow diet (CD) and filtered water ad libitum for 30 days, independently of sex.

UPF30: C57BL/6J mice receiving a diet enriched with ultra-processed foods (UPF) and a soft drink without gas ad libitum for 30 days, independently of sex.

### 2.3. Body Mass and Food Intake Assessment

The body mass evolution of each group was measured weekly, and food intake was assessed twice a week and expressed as a weekly average. The diet was changed two to three times a week, maintaining the sensory qualities of the pellets and foods. We determined the amount of food ingested by weighing the difference between the food offered and the remaining amount. We calculated energy consumption by using the energy content of each food, expressed in kilocalories per gram (kcal/g).

### 2.4. Fasting Blood Glucose, Insulin Levels, and HOMA Calculation

At 10 and 30 days of age, mice fasted for 12 h underwent blood collection for metabolic assessment. Blood glucose was measured from tail vein samples using an Accu-Chek Guide glucometer (Roche Diagnostics, Mannheim, Germany). For insulin determination, blood was collected either from the tail with heparinized capillaries (day 10) or by whole-body decapitation into heparin-coated tubes (day 30). Samples were centrifuged at 3000 rpm for 15 min at 4 °C to obtain plasma. Plasma insulin concentrations were quantified using an ELISA kit (cat. #EZRMI-13K, Merck, Darmstadt, Hesse, Germany). Insulin resistance was estimated using the homeostasis model assessment (HOMA-IR), calculated as [fasting glucose (mmol/L) × fasting insulin (µU/mL)]/22.5 [31].

### 2.5. Glucose Tolerance Test (GTT) and Insulin Tolerance Test (ITT) in Awake Mice

We performed an intraperitoneal (IP) glucose tolerance test (GTT) after a 6 h fast, starting at 8 a.m. At 2 p.m., we determined baseline glucose levels and administered a glucose solution (2.0 g/kg) IP. Subsequently, we collected blood from the tip of the tail over a 2 h period at 15, 30, 60, and 120 min to determine blood glucose levels in response to the IP injection.

We performed an intraperitoneal (IP) insulin tolerance test (ITT) after a 4 h fast, starting at 8 a.m. At 12 a.m., we determined baseline glucose levels and administered a regular insulin solution (0.75 IU/kg) IP. We chose the insulin dose based on previous studies demonstrating adequate glucose lowering without causing severe hypoglycemia [32,33]. Mice were conscious during the procedure. Subsequently, we collected blood from the tip of the tail over a 1 h period at 15, 30, and 60 min to determine blood glucose levels in response to the IP injection.

### 2.6. Pro- and Anti-Inflammatory Cytokine Concentrations

We anesthetized the mice by intraperitoneal injection of a mixture containing ketamine (300 mg/kg) and xylazine (30 mg/kg). The loss of pedal and corneal reflexes was used as a control for anesthesia. Subsequently, we euthanized the mice by decapitation.

Blood samples were collected during euthanasia. The mice had previously fasted for 12 h, and all samples were centrifuged for 20 min at 3500 rpm at 4 °C in a Universal centrifuge (320 R Hettich Zentrifugen, Darmstadt, Germany) to separate the serum. After collecting the serum into a separate tube, the samples were stored at −20 °C for future analysis.

We used the MILLIPLEX^®^ multiplex cytokine assays (MCYT1-190K, Merck Millipore, Darmstadt, Germany) to quantify the concentrations of pro- and anti-inflammatory cytokines—TNF-α, IL-1β, IL-6, and IL-10—in 25 μL of the same serum sample. According to the manufacturer, the minimum detectable concentrations (pg/mL) for these analytes using the overnight protocol were TNF-α (2.07), IL-1β (6.61), IL-6 (2.49), and IL-10 (4.88). Samples were run in duplicate and analyzed with the Luminex X-MAP platform at the Central Laboratory of Technologies for High-Performance Life Sciences (LACTAD), State University of Campinas (UNICAMP).

### 2.7. RNA Extraction and Real-Time PCR

Total RNA from the intestine was isolated using Trizol reagent (Thermo Fisher, Waltham, MA, USA) following the manufacturer’s instructions. The proximal colon was collected for total RNA extraction using Trizol^®^ (Thermo Fisher, Waltham, MA, USA).

Samples containing 1 μg of RNA were subjected to reverse transcription (RT) using random primers, DTT (100 mM) (Invitrogen, Carlsbad, CA, USA), dNTP mix (10 mM) (Applied Biosystems, Vilnius, Lithuania), and the SuperScript II enzyme (200 U) (Invitrogen, Carlsbad, CA, USA). PCR reactions were performed in a final volume of 10 μL, consisting of 12.5 ng of cDNA, 10 pM of specific primers, and Sybr Green PCR Master Mix (Thermo Fisher, Vilnus, Lithuania). Results were detected using the 7500 real-time PCR system (Applied Biosystems, Foster City, CA, USA).

The primers used were: Tumoral necrosis factor α (TNFα): F: CCCTCACACTCAGATCATCTTCT; R: GCTACGACGTGGGCTACAG

Interleukin 6 (IL6): F: CACGGCCTTCCCTACTTCAC; R: GGTCTGTTGGGAGTGG-TATC

Interleukin 10 (IL-10): F: GCTCTTACTGACTGGCATGAG; R: CGCAGCTCTAGGAG-CATGTG

Mucin 1 (Muc1): F: GCAGTCCTCAGTGGCACCTC; R: CAC-CGTGGGCTACTGGAG

Mucin 2 (Muc2): F: ACAAAAACCCCAGCAACAAG; R: GAG-CAAGGGACTCTGGTCTG

Cadherin (Cad): F: CCTGTCTTCAACCCAA GCAC; R: CAACAACGAACTGCTGGTCA

Occludin (Occld): F: CTCTCAGCCAGCGTACTCTT; R: CTCCATAGCCACCTCCGTAG

Interleukin 1β (IL1β): F: GCAACTGTTCCTGAACTCAACT; R: ATCTTTT-GGGGTCCGTCAACT

Ribosomal protein L32 (Rpl32): F: CAAAATCGCCCTATTCCTCA; R: AGACCCAGCTTCGTTCTCCT

The relative mRNA levels were determined after normalization to Rpl32 using the ΔΔCT method.

### 2.8. Collection of Feces for DNA Extraction

After euthanasia, we made a longitudinal incision in the mice and collected the colon and feces from the distal portion of the intestine. The feces and colon samples were immediately frozen in liquid nitrogen (−80 °C) until further processing.

### 2.9. Metagenomic Characterization

We extracted bacterial DNA using the ZymoBIOMICS DNA Miniprep Kit (Zymo Research, Irvine, CA, USA, ref: D4300), following the manufacturer’s instructions. We quantified the DNA before sequencing and stored the samples at −20 °C until further molecular analysis was performed.

Next, we prepared the amplicon library and amplified the target marker gene. We amplified the V3–V4 region of the bacterial 16S rRNA gene (~470 bp) using primers 341F (5′-CCTAYGGGRBGCASCAG-3′) and 806R (5′-GGACTACNNGGGTATCTAAT-3′). Sequencing was performed on the Illumina PE 250 platform. After sequencing, we mapped the reads to the reference database specified in the QIIME2 documentation, removing low-quality and chimeric reads during the processing.

For functional prediction, we used PICRUSt2 (version 2.4.2) with default settings. PICRUSt2 employs phylogenetic placement of ASVs and has been validated against metagenomic data [34]. We did not perform any direct validation in this study.

Finally, we used Python (version 3.7) for data visualization and exploration of biological data.

### 2.10. Statistical Analysis

We expressed the results as means ± SEM. We compared the male and female groups separately to assess the effect of the diets on the variables studied. First, we checked for normality using the Shapiro–Wilk test. Based on the results, we applied parametric tests (two-tailed unpaired Student’s *t*-test and Two-Way ANOVA with Bonferroni post hoc) or nonparametric tests (Mann–Whitney test). We corrected multiple comparisons with Bonferroni’s post hoc test, as indicated in the figure legends.

All analyses were performed using GraphPad Prism version 10.4.0, and a *p*-value less than 0.05 was considered statistically significant. Detailed statistical information is available in the figure legends.

## 3. Results

### 3.1. UPF Diet Alters Food Intake, Adiposity, and Glucose Metabolism

We recorded body mass (in grams) weekly during the 10-day intervention period. We did not observe a significant difference in this parameter over the 10 days between diets in male mice. The female’s body mass was also similar between the diets of the 10-day intervention (Figure 1A). Despite no changes in body mass, we observed a significant increase in gonadal adipose tissue compared to controls after 10 days of the UPF diet only in females, not in males (Figure 1B). Regarding food intake, we observed that both males and females consumed more kilocalories when receiving a UPF diet for 10 days compared to a chow diet (Figure 1C). Blood glucose levels and insulin concentration after a 12 h overnight fast were significantly decreased in males on the UPF diet compared to the chow diet. For the females, we observed lower blood glucose levels after fasting under the UPF diet compared to the chow diet. Insulin concentration levels were similar between diets in this sex (Figure 1D,E). The HOMA-IR indicates that both males and females were more insulin-sensitive on a UPF diet for 10 days than those of the same sex receiving a chow diet (Figure 1F).

After 30 days, the UPF-rich diet did not significantly change the body mass of males and females. However, the gonadal adiposity in both genders, which received the UPF, was remarkably higher than in the control animals (Figure 1G,H). Regarding feeding, we observed that both males and females consumed more kilocalories when receiving a UPF diet for 30 days compared to a chow diet (Figure 1I).

Blood glucose levels after a 12 h overnight fast were significantly decreased in females, but not in males, on the UPF diet compared to the chow diet (Figure 1J). We did not observe changes in insulin concentration levels after fasting under the UPF diet compared to the chow diet in both sexes (Figure 1K). We did not observe a significant difference in the HOMA-IR between males and females on a UPF diet for 30 days compared to those of the same sex receiving a chow diet (Figure 1L).

### 3.2. UPF Alters Insulin and Glucose Tolerance Tests

Although blood glucose levels were higher before insulin injection in males and females fed UPF for 10 days, the glycemic response to insulin during the ITT was not statistically different between the diets of either sex. At 60 min, blood glucose levels were lower in males, not females, under the UPF diet (Figure 2A,B).

The GTT showed that only females, not males, fed a UPF for 10 days, were less efficient at metabolizing glucose than the control group. Especially 15 min after glucose injection, the glycemic response was higher than that of the female control (Figure 2C,D).

Similar to 10 days of the UPF diet, blood glucose levels were higher before insulin injection in males and females fed a UPF for 30 days. However, the glycemic response to insulin during the ITT was not statistically different between the diets of either sex (Figure 2E,F).

Males fed a UPF or control diet for 30 days demonstrated similar glucose tolerance during the GTT. In contrast, females fed a UPF for 30 days demonstrated less efficiency in metabolizing glucose than female controls (Figure 2G,H).

### 3.3. Results from a Pilot Experiment in Adult Male Mice After 13 Weeks of a UPF

In a separate pilot experiment that we had previously developed, specifically in adult male mice after 13 weeks of a UPF, we observed a significant increase in body mass in the 13th week (Appendix A). Similarly, the UPF animals had increased epididymal adipose tissue (Appendix A). Because white adipose tissue secretes leptin in proportion to its mass (Pan and Myers 2018 [35]), we observed that serum leptin levels increased in the group of animals fed the UPF diet compared to the CD group (Appendix A). We also found consistent results in food intake, with higher kilocalorie intake in the UPF group (Appendix A).

The GTT results indicate glucose intolerance in male mice subjected to a UPF diet for 13 weeks in the pilot experiment, which exhibited altered glycemic responses at 30, 45, and 60 min following glucose administration (Appendix A), indicating impaired glucose tolerance. Serum LPS levels were higher in the UPF group (Appendix A).

### 3.4. UPF Altered Serum Pro-And Anti-Inflammatory Cytokine Levels After 30 Days, Depending on the Sex

After 10 days of receiving a UPF diet, serum pro- and anti-inflammatory cytokine levels were similar to those of a control diet, independent of sex (Figure 3A,B). However, after 30 days on the UPF diet, males exhibited a significant increase in serum IL-6 levels compared to CD (Figure 3C), and no changes in the TNF-α, IL-1β, and IL-10 levels (Figure 3C). In contrast, in females, despite similar pro-inflammatory levels of TNF-α, IL-6, and IL-1β, we detected a decrease in IL-10 levels after 30 days of UPF diet compared to controls (Figure 3D). This result suggests that a 30-day UPF diet may modulate pro- and anti-inflammatory markers differently in females. While in males, we observed an increase in serum IL-6, a pro-inflammatory cytokine, in females, we found a decrease in serum IL-10, a marker of anti-inflammation.

### 3.5. UPF for 10 Days Altered the Expression of Intestinal Barrier Markers and Inflammatory Cytokines Depending on the Sex

To investigate whether a UPF diet might alter the intestinal barrier and local inflammatory state, we performed RT-PCR on the intestinal tissues of male and female mice after 10 days of receiving a UPF or a chow diet. We did not observe changes in *TNF-α*, *IL-6*, *IL-1β*, and *IL-10* expression in the gut of males after a 10-day UPF diet (Figure 4A). Similarly, the expression of *E-cadherin*, *mucin 1* and *2*, and *occludin* did not differ in the gut of male mice between diets (Figure 4B).

We did not observe changes in the expression of cytokines, including *TNF-α*, *IL-1β*, *IL-6*, and *IL-10*, after a UPF diet for 10 days in females (Figure 4C). Regarding the intestinal barrier, we did not observe significant differences in *E-cadherin*, *mucin 1* and *2*, and *occludin* gut expression in females after 10 days of a UPF diet (Figure 4D).

### 3.6. UPF for 30 Days Altered the Expression of Intestinal Barrier Markers and Inflammatory Cytokines

Next, we investigated whether a prolonged diet might impair intestinal barrier regulation and inflammatory profile after 30 days of receiving UPF or a chow diet. We did not observe changes in cytokine expression in the gut of males after 30 days of a UPF diet (Figure 4E). However, we observed a decrease in *E-cadherin* and *occludin* expression in the gut of male mice. *Mucin 1* and 2 expressions did not change between groups (Figure 4F). In female mice, we found elevated *TNF-α* expression in the gut after 30 days of receiving the UPF diet. However, the expression of *IL-6*, *IL-1β*, and *IL-10* in the gut was not different between the diets (Figure 4G). Despite higher *TNF-α* expression in the gut, we did not detect significant changes in *E-cadherin*, *mucin 1* and *2*, and *occludin* expression in females after 30 days of receiving the UPF diet (Figure 4H).

### 3.7. UPF Diet for 30 Days Alters the Gut Microbiota of Male and Female Mice Differently

We analyzed the cecal content from male and female mice exposed to the UPF and control diet for 30 days, using Illumina PE 250 for sequencing.

In the present study, females showed a significant reduction in the relative abundance of Bacteroidetes compared to males, with the UPF group exhibiting a lower percentage (Figure 5A). Additionally, females showed an increased abundance of Actinobacteriota and Verrucomicrobiota.

Alpha diversity, which reflects species richness and overall microbial diversity within a sample, showed no significant differences between groups, as assessed by the Shannon index. However, the Simpson index indicated that females displayed more diversity after the UPF diet than the control diet (Figure 5C,D).

The gut microbial community structure across groups was analyzed using PCoA based on Bray–Curtis dissimilarity, with statistical significance determined via PERMANOVA, comparing the true F-statistic against 999 randomly permuted F-statistics. We demonstrated a partial separation between the two diets (UPF × CD): for male groups, along axes 1 and 2, and for female groups, along axes 1 and 3. The general comparison revealed a notable difference among all groups (Figure 5E).

### 3.8. Genus-Level Classification

At the genus level, we identified 30 taxa. In Male UPF30, we demonstrated significant alterations in Faecalibaculum, Roseburia, Ruminococcus, Muribaculaceae, Parabacteroides, Alistipes, Bacteroides, Bifidobacterium, Enterohabidus, and Desulfovibrio (Table 1). In contrast, Female UPF30 exhibited significant changes in Dubosiella, Ruminococcus, GCA-900066575, Prevotellaceae, and Bifidobacterium (Table 2).

In Male UPF30, the relative abundance of Bacteroides, Parabacteroides, Faecalibaculum, Bifidobacterium, Alistipes, and Desulfovibrio increased, while Roseburia, Muribaculacea, and Enterohabidus decreased.

Both Male and Female UPF30 exhibited a reduction in *Ruminococcus* and an increase in *Bifidobacterium*. Additionally, in Female UPF30, *GCA-900066575* and *Dubosiella* increased, while *Prevotellaceae* reduced.

These findings suggest a possible influence of UPF on microbiota composition, warranting further investigation.

## 4. Discussion

Our study demonstrates that a diet rich in ultra-processed foods has an impact on energy metabolism. The course of exposure to the diet also revealed significant differences in glucose metabolism and insulin sensitivity. The induction of adiposity, inflammation, and dysbiosis, as well as changes in genes related to intestinal permeability, differed in males and females.

It was evident that the UPF diet resulted in higher caloric intake in both females and males, starting 10 days after its implementation. This result is consistent with previous data published by our group and others [28,36]. After 10 days of a Western diet, there was a decrease in insulin signaling and protein kinase B (Akt) phosphorylation in the hypothalamus associated with an impairment of the anorexigenic effect of insulin, leading to hyperphagia [28].

Increased energy consumption may result in increased gonadal fat mass in both sexes, particularly after 30 days. The increased fat mass observed aligns with previous studies, which have shown that consuming highly palatable diets can lead to an increase in adipose tissue mass [28,37,38]. Highly palatable diets can also increase thermogenesis and overall energy expenditure (EE) [39,40]. Although we did not perform an energy expenditure analysis in this study, in males receiving the UPF diet, the elevation of EE might have contributed to alleviating the increase in adiposity in this group after 10 days of diet. Nevertheless, such increases in energy expenditure may not be sufficient to offset excessive caloric intake, at least when the diet persists for more extended periods, thereby still promoting adiposity in both sexes after 30 days [39,40]. The elevated serum leptin levels and higher gonadal fat mass in our pilot study in males support the notion that a UPF diet increases adiposity when administered for an extended period.

To our surprise, animals fed a UPF diet for 10 days exhibited significantly lower fasting blood glucose levels and a lower HOMA-IR, suggesting a more insulin-sensitive state in both males and females. We did not observe significant differences in the ITT after 10 days of UPF, likely due to the varying fasting conditions. Mice were fasted for 4 h in the morning before the ITT experiment began. Distinct from the blood collection made to determine fasting levels of blood glucose and insulin for the HOMA calculation, which was a 10 h overnight fast.

Insulin levels generally correlate with adiposity; increased abdominal and visceral fat is associated with insulin resistance [41]. After 10 days of a UPF diet, males had lower fasting insulin levels, and this result was associated with no differences in gonadal fat mass. In females, we observed an increase in gonadal fat mass after 10 days of the UPF diet, with fasting insulinemia similar to that of the control group. However, after 30 days on a UPF diet, HOMA-IR was not significantly different between diets or sexes. Similarly, employing an ITT, we did not observe insulin resistance after 30 days of the UPF diet; this result was unexpected, as increased adiposity is typically associated with the induction of insulin resistance [42].

Although we did not observe changes in the GTT curve in males fed a UPF diet for 10 and 30 days, in our pilot experiment, male mice showed glucose intolerance when using the same diet for a prolonged time, 13 weeks. In another study, the authors found glucose intolerance in male rats after seven weeks of receiving a high-fat Western diet [43]. These findings combined suggest that, at least in rodent males, the disruption of glucose homeostasis may take time to develop. Compensatory mechanisms in the pancreas, such as an increase in β-cell mass and insulin secretion, require a more extended period for development [44,45] and may be implicated in the results we found.

To mitigate the impact of adiposity on metabolic changes, previous studies using palatable or cafeteria-style diets have included pair-fed groups [40,46]. In our study, the absence of a pair-fed control group is an important limitation, making it challenging to distinguish the effects of caloric load from those of UPF on energy intake and adiposity.

Both increased adiposity and hyperphagia significantly contribute to altering energy metabolism and triggering inflammatory responses. Considering that we observed hyperphagia in all groups fed a high-UPF diet, regardless of sex and duration of dieting, pair-fed groups would be of great insight.

However, regarding thermogenesis, LeBlanc et al. (1997) observed an increase in diet-induced thermogenesis and a reduction in feed efficiency (body weight gain/100 kJ intake) in rats under a cafeteria diet ad libitum and pair-fed with control rats [40].

In the gut microbiota, rats fed a cafeteria diet, which weighed the same as control rats on a chow diet, had a Firmicutes/Bacteroidetes ratio and the abundance of typical families and genera similar to those of rats fed a cafeteria diet ad libitum, not to those of the control rats [47]. In humans, the effects of diet have a more significant impact on the composition of the gut microbiota [48,49].

Collectively, these studies suggest that, in terms of diet-induced thermogenesis and gut microbiota composition, the nature of the diet outweighs the effects of caloric load and adiposity.

The females developed glucose intolerance after 10 to 30 days of receiving a UPF diet. These findings suggest that females are more prone to UPF effect on glucose homeostasis, and males require a longer period on a high-calorie diet to experience the same endpoints.

The phenotypic differences between the sexes we observed above might be due to the distinct regulation of the immune system function in males and females. Females have more severe immune responses than males, and most autoimmune diseases tend to be more prevalent in females [50]. A considerable part of the difference between the sexes is attributed to estrogen (and progestin) levels and responses [51]; however, we cannot rule out the multifactorial nature of this effect. Human and rodent experimental models indicate a direct correlation between the onset and severity of multiple autoimmune diseases and changes in circulating estrogen levels [52]. Consistent with this knowledge, in our study, females showed increased intestinal expression of IL-1β and TNF-α after 10 and 30 days of receiving the UPF diet. Gil-Cardoso et al. (2017) also observed increased expression of TNF-α mRNA in the ileum of Wistar rats fed a highly palatable diet for 14 weeks [53]. Since estrogen receptors are present in male and female intestinal epithelial cells [54], the absence of pro-inflammatory cytokine expression in the intestine of male mice may reflect exposure to different estrogen concentrations.

Estrogen plays a central role in diet-induced metabolic regulation [55,56]. Evidence indicates that estrogen receptor alpha (Erα) activity plays a key role in protecting females from metabolic dysfunctions induced by a high-fat diet. The protection of the estrogen/ERα influences lipid accumulation, insulin signaling, energy expenditure, and thermogenesis, contributing to female metabolic resilience [56]. Additionally, under a cafeteria (Western-type) diet, hepatocyte-specific ERα is essential for reverse cholesterol transport and for preventing lipid accumulation and atherosclerotic lesion formation, particularly in females [55]. Together, these findings suggest that estrogen signaling influences sex-specific metabolic responses, depending on the type of dietary challenge.

Females are more susceptible to the metabolic impacts of hormonal fluctuations. Estradiol and progesterone have been positively associated with insulin resistance [57]. Studies in women with type 1 diabetes have shown increased blood glucose levels, greater glycemic variability, and higher insulin requirements during the luteal phase [58]. In mice, the classical model for type 1 diabetes is the female NOD, as male NOD mice are generally not used as experimental groups due to their naturally lower basal incidence of T1D compared with females [59,60].

Although the observed patterns suggest that UPF consumption has sex-dependent effects, we did not perform a formal interaction analysis (diet × sex). Further studies are needed to determine whether UPF components can alter estrogen production or receptor signaling in the gut in this context.

Males on a UPF diet for 30 days displayed a reduction in the intestinal *occludin* and *E-cadherin* gene expression, suggesting an impairment of a few gut TJ and AJ markers, respectively. Although we did not observe changes in intestinal cytokine expression, the increased serum IL-6 levels may be associated with altered intestinal permeability, as elevated IL-6 has been previously linked to disruption of the intestinal barrier function [61]. However, the link between increased IL-6 levels and intestinal permeability in our study is merely speculative, as it lacks direct intestinal permeability assays. Gil-Cardoso et al. (2017) [53] found changes in the expression of intestinal permeability genes in female Wistar rats fed a cafeteria diet for 17 weeks, while inflammatory genes appeared early, at 14 weeks. Collectively, more definitive experiments measuring gut permeability, such as FITC-dextran and serum zonulin, are needed to confirm the effect of the UPF diet on the expression of TJ and AJ genes in the intestine over 30 days.

Serum LPS levels are an indirect measurement of intestinal permeability [62]. After high-fat exposure, LPS enters the bloodstream via leakage across the intestinal epithelial cell barrier via the transcellular pathway. Obese individuals exhibit elevated levels of LPS [63]. In our pilot experiment, we observed a significant increase in serum LPS levels in the UPF group of males compared to the CD group. This result may be attributed to dietary fat in the UPF group, changes in intestinal barrier function, or the possible translocation of LPS from Gram-negative bacteria in the intestine. Additional studies are necessary to explore these issues in detail.

At the phylum level, females exposed to the UPF diet showed a significant reduction in Bacteroidetes abundance, accompanied by an increase in Actinobacteria and Verrucomicrobiota abundance. These changes are consistent with prior findings linking UPF and high-fat diets to dysbiosis and altered metabolic flexibility [64,65]. We also found that the Simpson index decreased in females, suggesting higher microbial diversity due to the dominance of specific taxa, even though the Shannon index stayed unchanged in females [66].

The UPF diet resulted in distinct alterations in male and female mice at the genus level. Many of these changes point to a pro-inflammatory profile. In male UPF, *Desulfovibrio* and *Bacteroides* were increased; these bacteria are known to produce LPS from their cell walls and hydrogen sulfide, which contribute to intestinal barrier disruption and systemic inflammation [67,68]. Reducing *Roseburia* and *Muribaculaceae*, both SCFA-producing genera, may compromise gut epithelial integrity and anti-inflammatory signaling [69]. Collectively, these results might contribute to the impairment of barrier genes found in males.

Although *Faecalibaculum* increased—it is also a butyrate producer—its metabolic contribution is less potent compared to *Roseburia*’s, and some studies suggest that inflammatory conditions might elevate it [70].

In females, the UPF diet led to an increase in *Dubosiella*, *GCA-900066575*, and Bifidobacterium, as well as a decrease in *Prevotellaceae* and *Ruminococcus*.

A high-fat diet-induced obesity model may have increased *Dubosiella* in the intestines, suggesting that an elevation in this bacterium’s abundance participates in intestinal dysbiosis [71]. *Roseburia* is a butyrate-producing genus with anti-inflammatory properties that supports gut health; its decrease in our study may promote a pro-inflammatory state and disrupt metabolic balance [72].

Reducing *Prevotellaceae* and *Ruminococcus*, as observed in the females on a UPF diet, may imply a loss of fiber degradation capacity and SCFA production, potentially influencing glucose metabolism and mucosal health [73]. Interestingly, both sexes showed reduced *Ruminococcus*, suggesting some shared microbiota responses to UPF despite sex-specific patterns. Another unexpected response we observed in both sexes was the increased *Bifidobacterium*, which is often considered healthful; in this context, we can speculate that it may be a compensatory response or microbial imbalance. However, a deeper investigation is needed to confirm the role of increased *Bifidobacterium* after 30 days of a UPF diet [74].

Principal coordinates analysis (PCoA) revealed notable alterations in the structure of microbial communities among the groups, with partial separation influenced by both sex and dietary factors. These findings support the notion that the consumption of ultra-processed food (UPF) drives microbiota restructuring in a sex-specific manner, which may help explain the sex-based metabolic differences observed.

Although female mice generally exhibit improved glucose metabolism, even in response to high-fat diets (HFD) [75,76,77], few studies have assessed the impact of ultra-processed food (UPF)-rich diets, which contain additional components beyond fat. Additives commonly found in UPFs can alter gut microbiota composition and function, including reduced diversity, increased pro-inflammatory taxa, and changes in microbial metabolite production, which may directly modulate glucose metabolism [78]. Supporting this, an in vitro study simulating the human gut microbial environment (M-SHINE) demonstrated that additives induce changes in microbial gene expression and community composition that impact glucose metabolism [79].

Further studies are necessary to elucidate the mechanistic links between diet-driven alterations in microbiota and the observed metabolic and immunological outcomes. To assess the mechanism involved in microbiota changes, several assays can be considered, including bacterial localization relative to the intestinal mucosal surface using fluorescent in situ hybridization (FISH) and quantification of IgA-coated bacteria [80]. Additionally, to confirm the isolated impact of microbiota on the observed changes, fecal microbiota transplantation would be a valuable approach.

Finally, given the pronounced variation between males and females and the different phenotypes observed upon short and prolonged diet periods, we emphasize the importance of including both sexes and temporal studies under comparable experimental conditions.

## 5. Conclusions

Our results show that a diet high in ultra-processed foods (UPF) induced greater calorie intake as early as 10 days on a UPF diet. Fat accumulation occurs in both sexes, specifically after 30 days of exposure. The duration of UPF exposure influenced glucose metabolism, insulin sensitivity, inflammatory cytokine expression, gut barrier permeability gene markers, and microbiota composition. A ten-day UPF diet was associated with lower fasting blood glucose levels, without higher insulin levels, in both sexes. Females showed early impairment in glucose tolerance. Thirty days on a UPF resulted in more pronounced effects than 10 days in all parameters. Notably, male mice on UPF30 exhibited elevated systemic IL-6 levels, as well as reduced intestinal *occludin* and *E-cadherin* gene expression, suggesting inflammation and a potential for a leaky gut; however, direct permeability measurements are lacking. UPF30 also increased *TNF-α* expression in the gut and increased microbial diversity in females. Both sexes displayed dysbiosis, with females showing pronounced changes in the proportion between predominant phyla, and males showing only specific changes in bacterial genera.

## Figures and Tables

**Figure 1 nutrients-17-03116-f001:**
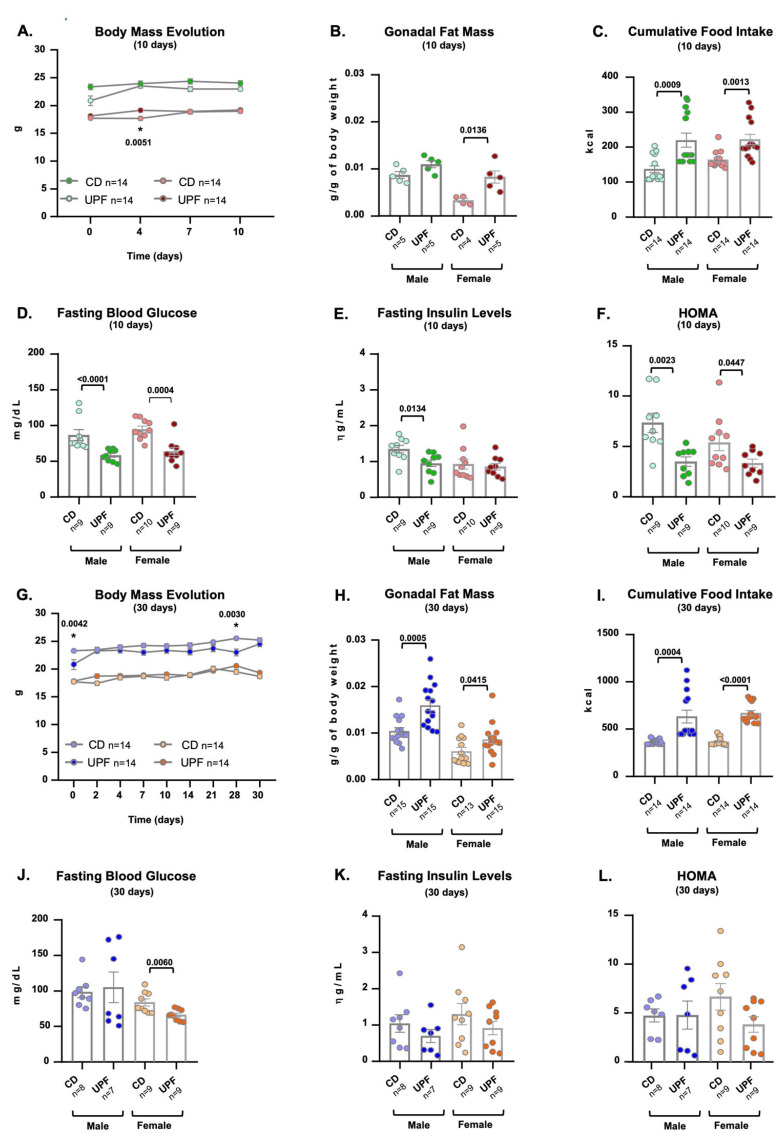
UPF diet alters adiposity and feeding. Male and female mice received a CD = chow diet or a UPF diet = ultra-processed food diet for 10 or 30 days, as indicated in the graphics. Body mass evolution (g) (**A**,**G**). Gonadal fat mass (g/g of body mass) (**B**,**H**). Cumulative food intake (kcal) (**C**,**I**). Fasting blood glucose (**D**,**J**). Fasting insulin levels (**E**,**K**). HOMA-IR (**F**,**L**). We expressed the results as means ± SEM. The number of animals is indicated in the figure. We compared the male and female groups separately to determine the effect of the diets on the variables studied. We applied a Two-way ANOVA with a post hoc test (Bonferroni) for Panels (**A**,**G**). Panels (**B**–**F**,**H**–**L**) were analyzed using the two-tailed Unpaired *t*-test. *p*  <  0.05 was considered statistically significant, and we indicated the value in the Figure. * indicates the corresponding *p*-value.

**Figure 2 nutrients-17-03116-f002:**
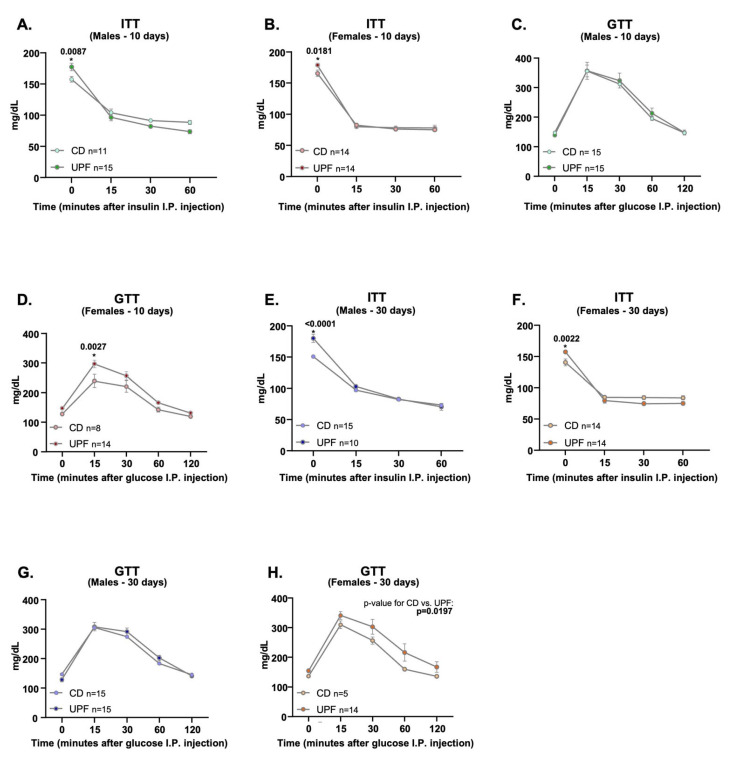
The effects of a UPF diet on insulin sensitivity and glucose metabolism. Male and female mice received a CD = chow diet or a UPF diet = ultra-processed food diet for 10 or 30 days, as indicated in the graphics. Insulin tolerance tests (ITT) in (**A**,**B**,**E**,**F**), glucose tolerance test (GTT) in (**C**,**D**,**G**,**H**). We expressed the results as means ± SEM. We compared the male and female groups separately to determine the effect of the diets on the variables studied. The number of animals is indicated in the figure. We applied a Two-way ANOVA with a post hoc test (Bonferroni) for all Panels. *p*  <  0.05 was considered statistically significant, and we indicated the value in the Figure. * indicates the corresponding *p*-value.

**Figure 3 nutrients-17-03116-f003:**
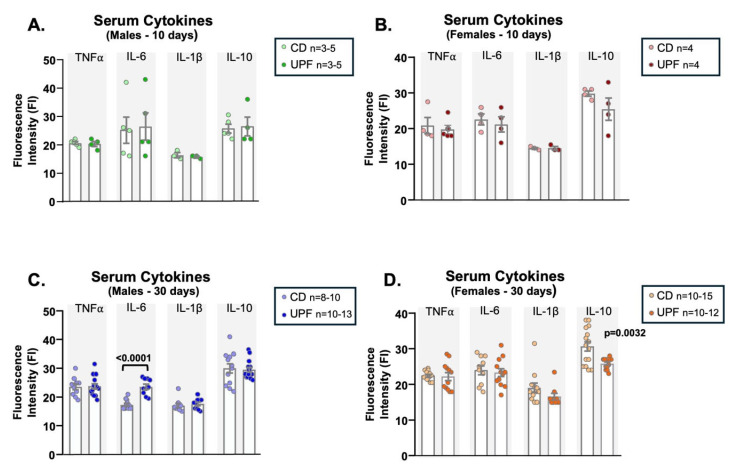
UPF diet alters serum cytokine levels. Serum cytokine levels from male and female mice after receiving a CD = chow diet or a UPF diet = ultra-processed food diet for 10 or 30 days, as indicated in the graphics. Serum cytokines (**A**–**D**). We expressed the results as means ± SEM. The number of animals is indicated in the figure. We compared the male and female groups separately to determine the effect of the diets on the variables studied. We analyzed Panels (**A**–**D**) using unpaired two-tailed *t*-tests, except for TNF-α in panel B, IL-6 and IL-10 in panel C, and IL-1β in panel D, for which we applied the nonparametric Mann–Whitney test. *p*  <  0.05 was considered statistically significant, and we indicated the value in the Figure.

**Figure 4 nutrients-17-03116-f004:**
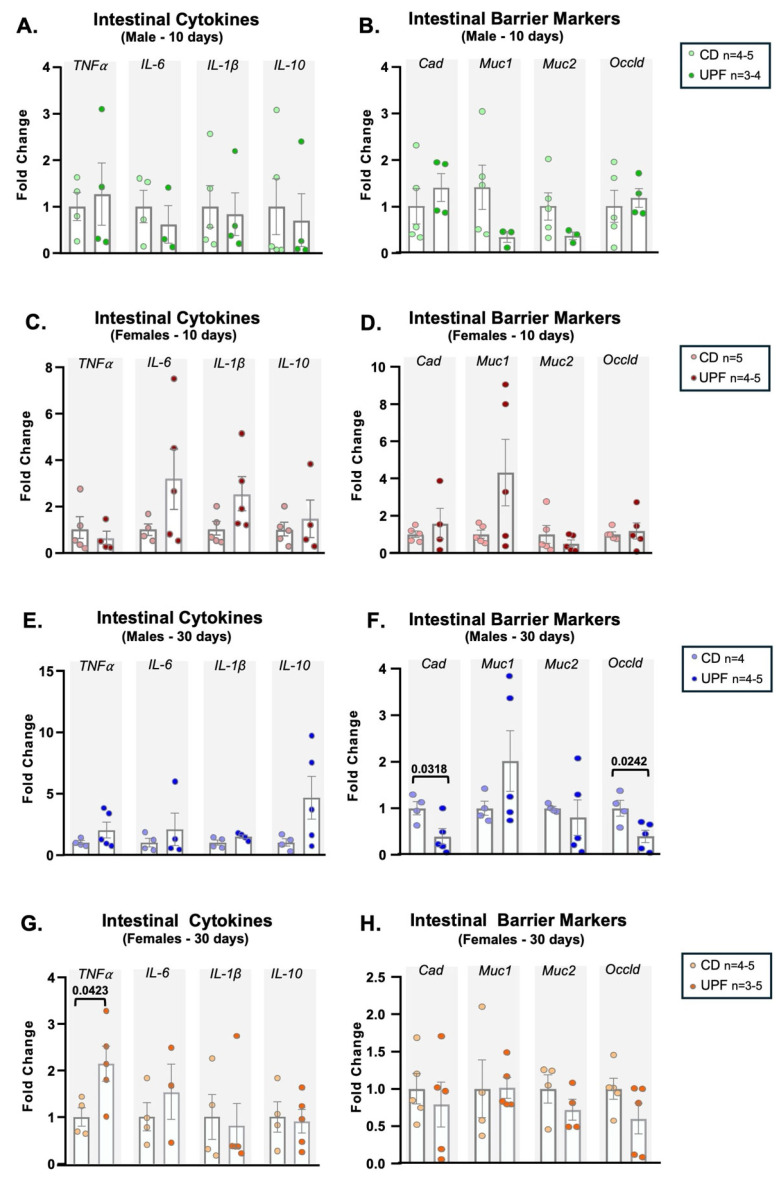
UPF diet affects intestinal barrier integrity and cytokine levels. Male and female mice received a CD = chow diet or a UPF diet = ultra-processed food diet for 10 or 30 days, as indicated in the graphics. Intestinal cytokines (**A**,**C**,**E**,**G**). Intestinal barrier markers (**B**,**D**,**F**,**H**). We expressed the results as means ± SEM. The number of animals is indicated in the figure. We compared the male and female groups separately to determine the effect of the diets on the variables studied. We analyzed Panels (**A**–**H**) using unpaired two-tailed *t*-tests, except for IL-10 in panel A, Muc1 in panel B, IL-6 in panel E, and IL-1β in panel G, for which we applied the nonparametric Mann–Whitney test. *p*  <  0.05 was considered statistically significant, and we indicated the value in the Figure.

**Figure 5 nutrients-17-03116-f005:**
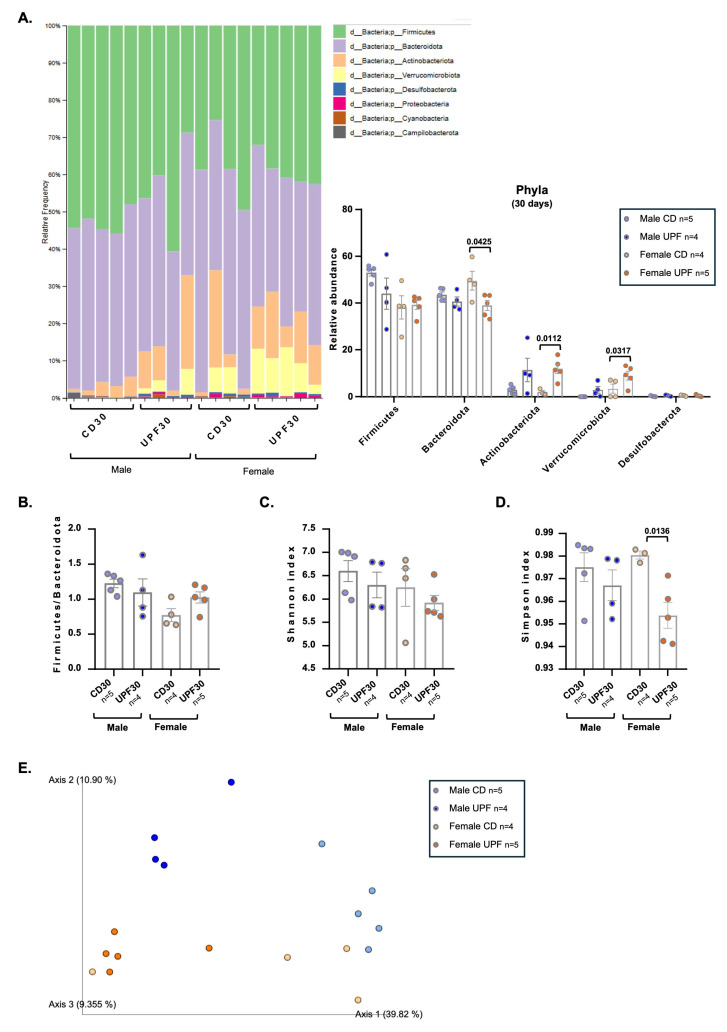
A UPF diet alters the composition of gut microbiota after 30 days. Male and female mice received a CD = chow diet or a UPF diet = ultra-processed food-enriched diet for 30 days, as indicated in the graphics. Phylogenetic relative abundance of fecal content (**A**). Firmicutes/Bacteroidetes ratio (**B**). Alpha diversity: Shannon Index (**C**) and Simpson Index (**D**) Principal coordinate analysis (PCoA) based on Bray–Curtis metrics (**E**). We expressed the results as means ± SEM. The number of animals is indicated in the figure. We compared the male and female groups separately to determine the effect of the diets on the variables studied. We analyzed Panels (**A**–**D**) using unpaired two-tailed *t*-tests, except for Actinobacteriota and Verrucomicrobiota in females (Panel A), for which we applied the nonparametric Mann–Whitney test. PERMANOVA was applied with a Bray–Curtis distance (*p* = 0.001) in Panel (**E**), for statistical significance. *p*  <  0.05 was considered statistically significant, and we indicated the value in the Figure.

**Table 1 nutrients-17-03116-t001:** Male genus relative frequency. Relative genus abundance in the fecal content of male mice after 30 days of UPF or CD feeding, n = 3–5 per group. UPF diet refers to the ultra-processed food diet, and CD refers to the control diet. We expressed the results as means ± SEM. We used the unpaired two-tailed *t*-tests to analyze data. * *p* <  0.05 was considered statistically significant.

Phylo	Genus	Relative Frequency		*p*-Value
		Male—CD30	Male—UPF30	
Firmicutes	*Dubosiella*	0.00 ± 0	8.20 ± 4.2	0.0637
	*Faecalibaculum*	0.15 ± 0.1	2.89 ± 0.7	0.0041 *
	*Lactobacillus*	7.50 ± 4.6	0.22 ± 0.1	0.2013
	*Roseburia*	1.83 ± 0.2	0.14 ± 0.2	0.0001 *
	*Blautia*	1.33 ± 0.5	0.26 ± 0.1	0.1123
	*Ruminococcus*	1.17 ± 0.3	0.25 ± 0.1	0.0272 *
	*GCA-900066575*	0.20 ± 0.1	0.32 ± 0.1	0.4105
Bacteroidota	*Muribaculacea*	50.30 ± 0.8	40.65 ± 2.1	0.0027 *
	*Muribaculum*	1.48 ± 0.3	1.39 ± 0.4	0.8573
	*Parabacteroides*	0.22 ± 0.1	2.09 ± 0.5	0.0082 *
	*Alistipes*	0.21 ± 0.1	0.86 ± 0.3	0.0323 *
	*Bacteroides*	0.46 ± 0.2	1.28 ± 0.01	0.0081 *
	*Prevotellaceae—UCG001*	0.26 ± 0.1	0.18 ± 0.1	0.4223
Actinobateriota	*Bifidobacterium*	0.00 ± 0	2.44 ± 0.7	0.0063 *
	*Enterohabidus*	3.18 ± 0.8	0.41 ± 0.1	0.0207 *
Verrucomicrobiota	*Akkermansia*	0.00 ± 0	3.18 ± 1.6	0.0622
Desulfobacterota	*Desulfovibrio*	0.13 ± 0.03	0.40 ± 0.1	0.0589 *

**Table 2 nutrients-17-03116-t002:** Female genus relative frequency. Relative genus abundance in the fecal content of male mice after 30 days of UPF or CD feeding, n = 3–5 per group. UPF diet refers to the ultra-processed food diet, and CD refers to the control diet. We expressed the results as means ± SEM. We used the unpaired two-tailed *t*-tests to analyze data. * *p* <  0.05 was considered statistically significant.

Phylo	Genus	Relative Frequency		*p*-Value
		Female—CD30	Female—UPF30	
Firmicutes	*Dubosiella*	0.00 ± 0	13.41 ± 4.0	0.0489 *
	*Faecalibaculum*	0.03 ± 0.02	2.08 ± 1.0	0.1642
	*Lactobacillus*	1.97 ± 0.9	0.11 ± 0.04	0.0928
	*Roseburia*	1.05 ± 0.7	0.20 ± 0.1	0.2117
	*Blautia*	0.53 ± 0.2	0.29 ± 0.1	0.3378
	*Ruminococcus*	0.75 ± 0.2	0.14 ± 0.03	0.0156 *
	*GCA-900066575*	0.11 ± 0.03	0.54 ± 0.2	0.0410 *
Bacteroidota	*Muribaculacea*	49.34 ± 4.3	39.93 ± 1.6	0.0615
	*Muribaculum*	3.07 ± 0.1	2.46 ± 0.3	0.231
	*Parabacteroides*	0.31 ± 0.03	1.08 ± 0.3	0.1126
	*Alistipes*	1.09 ± 0.3	0.76 ± 0.1	0.271
	*Bacteroides*	0.87 ± 0.4	0.18 ± 0.1	0.0735
	*Prevotellaceae—UCG001*	0.95 ± 0.4	0.02 ± 0.01	0.0301 *
Actinobateriota	*Bifidobacterium*	0.00 ± 0	5.58 ± 1.8	0.0566 *
	*Enterohabidus*	1.67 ± 0.6	0.32 ± 0.06	0.0654
Verrucomicrobiota	*Akkermansia*	3.85 ± 2.2	10.43 ± 2.1	0.0703
Desulfobacterota	*Desulfovibrio*	0.51 ± 0.1	0.43 ± 0.2	0.7497

## Data Availability

The raw data supporting the conclusions of this article will be made available by the authors on request.

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
