# Peer review of "Effects of Ultra-Processed Diets on Adiposity, Gut Barrier Integrity, Inflammation, and Microbiota in Male and Female Mice"

_nutrients, 2025, doi:10.3390/nu17193116_

Round 1
Reviewer 1 Report
Comments and Suggestions for Authors
The study provides valuable insights into sex-specific responses to a diet high in ultra-processed foods (UPF) in mice, an important and underexplored area. The authors effectively integrate multiple factors (adiposity, glucose intolerance, inflammation, dysbiosis) to explain the differential effects of UPF consumption between sexes. The data strengthen the findings by offering longer-term effects, particularly regarding adiposity and glucose metabolism. While the study contributes meaningful findings on the effects of UPF on metabolic and immune responses, it would benefit substantially from a major revision.
Major comments
1. The introduction provides a comprehensive overview, touching on relevant topics such as nutrition, metabolism, gut barrier integrity, and microbiota. However, it attempts to cover too many concepts without smooth transitions (e.g., the sudden shift from historical models to gut barrier function). Additionally, there are numerous grammatical issues and awkward phrasings, such as misplaced parentheses, inconsistent verb tenses, and sentence fragments. Furthermore, consistent terminology should be used: replace “gender” with “sex” throughout, as this refers to biological differences in mice, in alignment with your abstract.
2. Although sex differences are addressed, the data interpretation remains largely descriptive. The authors should more explicitly clarify whether interactions between diet and sex were statistically tested (e.g., using two-way ANOVA). Simply reporting separate male and female data is insufficient to support claims of sex-dependent effects. The interpretation could benefit from a deeper discussion of potential underlying molecular mechanisms. For instance, while estrogen is mentioned as a key player, further exploration of its role in diet-induced metabolic changes would strengthen the analysis.
3. Please clarify the rationale for selecting both 10- and 30-day durations. It would be helpful to justify whether these durations represent acute versus chronic exposure models, as this distinction would provide more clarity to readers.
4. The manuscript discusses "tendencies" (e.g., p = 0.0609, p = 0.0735) as though they support conclusions. While trends can be noted, they should not be interpreted as indicating biological effects without statistical significance. The authors should either tone down such interpretations or justify them with additional analysis (e.g., power analysis, sample size limitations).
5. UPF-fed mice consistently consumed more calories. The increased adiposity and metabolic changes could thus be attributed to hyperphagia rather than diet quality. Were pair-fed controls used or considered? Without this control, conclusions about UPF-induced metabolic or inflammatory changes should be qualified.
6. In the glucose metabolism section, the sentence “Males fed with UPF or a control diet” (lines 260–261) is abruptly cut off and unclear. Furthermore, the comparison between the 10- and 30-day results, as well as the pilot study data, is confusing and lacks a clear logical flow. The authors should make a clearer distinction between exploratory (pilot) and confirmatory (main) findings.
7. The microbiota analysis provides descriptive statistics, but the interpretation is underdeveloped. For example, what do changes in genera such as Dubosiella, Roseburia, or GCA-900066575 imply in terms of metabolic or inflammatory profiles? The authors should link these shifts in microbial taxa more directly to host phenotypes or known functional impacts.
8. While the authors mention that further studies are needed to understand the mechanisms driving the observed effects, they should outline specific experimental approaches for future research. This could include investigating how gut microbiota, inflammatory cytokines, and intestinal barrier function interact in response to UPF.
9. The increased food intake is associated with increased adiposity (lines 419–423), but the relationship between these factors requires further clarification. Is the increased adiposity purely a result of higher caloric intake, or do metabolic changes or altered energy expenditure contribute? This should be addressed to strengthen the paper’s conclusions.
10. The discussion section provides a detailed analysis of the findings, but it would benefit from greater conciseness. Certain topics are repeated or contain excessive background information. Streamlining the discussion of topics such as cytokine responses and microbiota alterations would allow for a more focused presentation of the study's key findings and their implications.
11. In lines 433–440, the authors mention insulin resistance and the ITT, which is intriguing. However, they should clarify whether the absence of insulin resistance could be due to compensatory mechanisms or whether the ITT is less sensitive to the early stages of insulin resistance. A more thorough discussion of why ITT was not sensitive in this model would enhance the understanding of this finding.
Minor Revisions
1. The statement "without affecting body weight" should be clarified. Did lean mass or water weight compensate for the lack of weight change?
2. Avoid implying causality too strongly in observational sections. For example, "caused inflammation" could be softened to "was associated with increased inflammatory markers."
3. Some cytokines are discussed in terms of gene expression (intestinal TNF-α), while others are discussed in terms of protein levels (serum IL-6). The authors should either standardize the terminology or clarify the distinction. Similarly, gene names should be formatted consistently (e.g., Muc2 vs. Muc 2) in line with accepted nomenclature.
4. The description of the UPF diet is detailed, but the proportions of its components (e.g., peanut, chocolate) are not specified. This limits reproducibility. Please provide precise percentages or weights for each component of the cafeteria diet.
5. In the methods section, specify the 16S rRNA region targeted, the primers used, and the DNA extraction and quantification methods prior to sequencing.
6. The section numbering skips from 2.8 to 2.10. Please correct this inconsistency. Additionally, there appear to be two sections labeled "3.5" in the results section, which should be revised for clarity.
7. Standardize the terminology for the diet ("chow diet" vs. "standard diet") throughout the manuscript to ensure consistency.
8. Several instances of awkward or incorrect phrasing (e.g., "intestine tissues," "e-cadherin," "significative differences") require careful proofreading for language accuracy.
Author Response
Dear reviewer,
Thank you for taking the time to review our manuscript and for providing us with all the valuable suggestions. This revision was an opportunity to improve the quality of our work.
We were able to run one more group to answer some of the questions; however, the number of animals and reagents was minimal.
We highlighted in blue all changes made in the manuscript.
We hope these revisions meet your expectations.
Best,
Patricia O. Prada and Clara M. Campolim

Reviewer 2 Report
Comments and Suggestions for Authors
Review of the manuscript: "Ultra-processed foods increase feeding and adiposity and alter gut barrier markers, inflammation, and microbiota: A Gender-Dependent Perspective"
# Global.
-The present manuscript investigates the impact of ultra-processed food (UPF) consumption on metabolic parameters, gut barrier integrity, inflammatory markers, and gut microbiota in male and female mice. The utilisation of both sexes, in conjunction with combined physiological and molecular analyses, and defined exposure periods (10 and 30 days), are evident strengths of the study. The manuscript's major strengths are as follows: firstly, it includes both male and female mice in order to explore sex-dependent responses; secondly, it employs a multi-dimensional approach covering physiology, cytokine profiling, gene expression, and microbiota composition; thirdly, it utilises well-selected acute (10-day) and chronic (30-day) exposure timelines.
-Nevertheless, the study should clarify methodology, strengthen statistical rigor, and moderate some interpretations before acceptance. It is evident that there is a paucity of detail concerning sample size estimation, randomisation, and blinding procedures. It is important to note the tendency to overinterpret correlational microbiota data as causal mechanisms. It is evident that there is a paucity of clarity with regard to statistical methodologies, in particular with respect to the implementation of multiple comparison corrections. Furthermore, there are numerous grammatical and formatting inconsistencies that have a detrimental effect on readability.
# Main Conceptual Errors and Recommendations
-The present study hypothesises that reliance on ITT/GTT is associated with insulin resistance. The manuscript relies on insulin tolerance tests (ITT) and glucose tolerance tests (GTT) to assess insulin sensitivity, yet these methods have limitations, and pilot clamp data appear contradictory. It is requested that hyperinsulinemic–euglycemic clamp studies be considered for performance, or alternatively that HOMA-IR calculations be provided, with the choice of insulin dose and test justified.
-Causality in Microbiota Findings. Current language suggests that fluctuations in bacterial genera directly precipitate inflammation; nevertheless, extant data merely demonstrate associations. It is imperative to reframe conclusions in a manner that accentuates associations over causation. In order to gain further mechanistic insight, the execution of follow-up fecal transplant experiments is recommended.
-Lack of Detailed Dietary Composition. The UPF diet is described by a macronutrient-based approach, encompassing percentages of nutrients without detailed breakdowns of individual components or assessments of palatability. It is imperative that a comprehensive table is included in the methods or supplemental material. This table must detail the macro- and micronutrient content of each UPF component. In addition, any sensory evaluation performed to confirm palatability must be described.
-Statistical Power and Sample Sizes. Several assays (e.g., qPCR, cytokine measurements) use small sample sizes (n = 3–5), which may lack sufficient power. Please, provide a priori power analyses to justify sample sizes, report confidence intervals, and consider increasing replicates where feasible.
#Analysis by lines
-Lines 1–4. Title. Consider shortening to “Sex-Dependent Effects of Ultra-Processed Diets on Adiposity, Gut Barrier Integrity, Inflammation, and Microbiota in Mice” to improve conciseness.
-Lines 15–18. The sentence “The consumption of highly palatable ultra-processed foods... increases the risk of morbidity and mortality” is vague. Please improve the sentence by specifying the diseases (e.g., obesity, type 2 diabetes) and, if possible, include quantitative risk information.
-Line 41. Please, fix the misplaced parentheses in “...industry adds both nutrients in the same product of UPF ((10).” It should read “(10)”.
-Lines 52–57. When referencing palatable diets in rats, please briefly explain how your mouse UPF model differs or extends these prior studies.
-Lines 60–63- Please, clarify the relationship between your definition of UPF and the NOVA classification (ref. 19), and explain why you adopt your in-house criteria.
- Lines 101–104. Describe how animals were randomized to diets and whether investigators were blinded during data collection.
-Line 118. While chow composition is detailed, UPF components lack specific macro- and micronutrient tables. Please, provide those details in the main text or supplementary materials.
-Line 136. Food intake was measured twice weekly but reported as a weekly sum. Clarify the calculation method and how day-to-day variability was addressed.
-Lines 143–146. Please, include details on fasting duration, whether mice were conscious or anaesthetised, the insulin dose rationale, and baseline stabilization times.
-Line 156. Please, provide catalogue numbers, assay sensitivity, and replicate numbers for the
-Lines 164–170. Please, list primer efficiencies, melting curve validation, and the stability of reference genes.
-Lines 192–198. Please, specify the PICRUSt2 database version, parameters for prediction, and any validation against metagenomic data.
-Lines 200–207. Please, define which parametric or nonparametric tests were applied to each dataset, how multiple comparisons were corrected, and the software (with version) used.
-Figure Legends- Replace broad “p < 0.05” labels with exact p-values and include the number of animals (n) per group for each panel.
-Lines 359–366 and 524–528. The linkage between IL‑6 increases and gut permeability is speculative without direct assays (e.g., FITC‑dextran). Please, rephrase to note associations and consider acknowledging the lack of direct permeability measurements.
# Other issues
-Use italics for gene symbols and plain text for proteins.
-Define all acronyms upon first use and ensure consistency (e.g., UPF vs. NCD vs. CD).
-Address grammatical inconsistencies throughout; a professional language edit is recommended.
Comments on the Quality of English LanguagePlease, see report
Author Response

(The authors gave the same response as above.)

Reviewer 3 Report
Comments and Suggestions for Authors
Description of a mouse model exploring the gender difference of a diet high in ultra-processed foods (UPF) consumption for 30 days. The time of the experiment was too short to induce significant carbohydrate metabolism disturbances, as a consequence of low adipose tissue accumulation. The authors showed dysbiosis in both genders and reduced microbial diversity in female mice. Contrary, only in male mice reduced intestinal occludin and e-cadherin gene expressions were shown.
The abstract is not informative.
The authors should explain in the introduction why they chose the mouth model for the experiment.
Is there any reason to show results after 10 days (or perhaps present them as supplementary).
The description of results should be shorter, and most results presented in the table. Only the most important figures. Adiposity changes may be combined with glucose metabolism. There is no need to describe statistical tests in the results section.
The discussion is long and difficult to follow. Please start from a short summary, like you did in the conclusion section. It is obvious that disturbances are more pronounced after 30 days of UPF.
Start from the changes in biota. Explain the potential mechanism of intestinal gene expression and present a hypothesis on why gender differences exist. Some of them are difficult to explain and should be further studied.
Finally, discuss the model in the context of the transition to humans.
Conclusions should be focused on the gender perspective. It is not clear why glucose intolerance developed earlier in female mice.
The sentence "Thus, our study demonstrates that UPF increases adiposity and food intake of mice in both genders, regardless of the exposure time." is simply not true.
Author Response

(The authors gave the same response as above.)

Round 2
Reviewer 1 Report
Comments and Suggestions for Authors
The authors have substantially improved the manuscript by addressing my concerns, including clarifying statistical analyses, refining the discussion of sex-specific effects, providing detailed methodological information, and enhancing clarity and language throughout. Their work provides novel, well-supported insights into how ultra-processed foods impact metabolism, inflammation, gut barrier function, and microbiota in a sex-dependent manner. The study is rigorous, comprehensive, and makes a valuable contribution to the field. I therefore recommend acceptance of the manuscript.
Author Response
Dear reviewer,
Thank you for your thoughtful evaluation of our manuscript and for your kind recommendation for acceptance. We greatly appreciate your constructive feedback throughout the review process, which allowed us to significantly improve the quality and clarity of our work.
Best regards,
All authors
Reviewer 2 Report
Comments and Suggestions for Authors
Review of the manuscript “Sex-Dependent Effects of Ultra-Processed Diets on Adiposity, Gut Barrier Integrity, Inflammation, and Microbiota in Mice”
nutrients-3635849
# The revised version of "Sex-Dependent Effects of Ultra-Processed Diets on Adiposity, Gut Barrier Integrity, Inflammation, and Microbiota in Mice" demonstrates significant enhancements compared to the previous draft. The introduction is now more concise and provides a clear rationale for the study, contextualising it with relevant epidemiological and mechanistic references. The methods section is well-structured and provides sufficient detail on the animal models used, the composition of the diets, the metabolic testing procedures and the molecular techniques employed. The results are systematically presented, and the addition of microbiota analysis significantly strengthens the translational value of the findings. The discussion situates the results within a broader scientific context, highlighting sex-specific differences and potential mechanistic pathways.
However, a few points should be considered before acceptance:
- While sex differences are emphasized, the absence of a formal diet × sex interaction analysis is a limitation. The authors acknowledge this, but at minimum, supplementary statistical analyses (or an explanation of why these were not feasible) should be included.
-The lack of a pair-fed control group makes it difficult to disentangle caloric load from UPF-specific effects. This limitation should be emphasized more clearly in the discussion.
- The conclusions about gut permeability rely on gene expression data without direct permeability assays (e.g., FITC-dextran, serum zonulin). The authors should either temper these claims or propose them explicitly as hypotheses requiring confirmation.
-Some unexpected results (e.g., increased Bifidobacterium abundance) are described as compensatory, but this remains speculative. More cautious language or references supporting this interpretation would improve credibility.
-Figures are generally clear, but some legends could be more self-explanatory (e.g., specify sample sizes for each sex in all panels). Providing raw data in a supplementary dataset would further enhance transparency.
---
In short, the manuscript represents a strong contribution to the field, with novel insights into sex-specific responses to UPF diets. After addressing the above points—particularly the statistical analyses and clarification of gut permeability claims—it should be suitable for publication.
Author Response
Dear Reviewer,
Thank you for the thorough evaluation of our revised manuscript and for recognizing the improvements made. We appreciate your constructive feedback and detailed suggestions, which have been very valuable in further strengthening our work. Below, we provide a point-by-point response to each of your comments and have made the corresponding revisions to the manuscript.
Best regards,
All authors

Reviewer 3 Report
Comments and Suggestions for Authors
The paper was improved. It can still benefit from a shorter description of the results. Discussion is much more clear.
Author Response
Dear Reviewer,
We appreciate your thoughtful feedback. We are submitting a revised version, which we believe has been significantly improved after incorporating the suggestions from all reviewers.
Best regards,
All authors